# Tumor Location Impacts the Development of Radiation Necrosis in Benign Intracranial Tumors

**DOI:** 10.3390/cancers15194760

**Published:** 2023-09-28

**Authors:** Matthias Demetz, Julian Mangesius, Aleksandrs Krigers, Meinhard Nevinny-Stickel, Claudius Thomé, Christian F. Freyschlag, Johannes Kerschbaumer

**Affiliations:** 1Department of Neurosurgery, Medical University of Innsbruck, Anichstr. 35, 6020 Innsbruck, Austria; 2Department of Radiation Oncology, Medical University of Innsbruck, 6020 Innsbruck, Austria

**Keywords:** skull base, radiosurgery, radiation necrosis, neuro-oncology

## Abstract

**Simple Summary:**

The impact of tumor location on the development of radiation necrosis is still unclear. We evaluated 205 patients with benign intracranial tumors, who underwent stereotactic radiosurgery. A total of 15.6% developed radiation necrosis after a median of 10 months. According to our data, tumors located at the skull base showed a significantly lower risk of radiation necrosis (HR 0.252, *p* < 0.001). The data from this study suggest that the tumor location at the skull base can affect the development of radiation necrosis in benign intracranial neoplasms.

**Abstract:**

Background: Radiation necrosis (RN) is a possible late complication of stereotactic radiosurgery (SRS), but only a few risk factors are known. The aim of this study was to assess tumor location in correlation to the development of radiation necrosis for skull base (SB) and non-skull base tumors. Methods: All patients treated with radiosurgery for benign neoplasms (2004–2020) were retrospectively evaluated. The clinical, imaging and medication data were obtained and the largest axial tumor diameter was determined using MRI scans in T1-weighted imaging with gadolinium. The diagnosis of RN was established using imaging parameters. Patients with tumors located at the skull base were compared to patients with tumors in non-skull base locations. Results: 205 patients could be included. Overall, 157 tumors (76.6%) were located at the SB and compared to 48 (23.4%) non-SB tumors. Among SB tumors, the most common were vestibular schwannomas (125 cases) and meningiomas (21 cases). In total, 32 (15.6%) patients developed RN after a median of 10 (IqR 5–12) months. Moreover, 62 patients (30.2%) had already undergone at least one surgical resection. In multivariate Cox regression, SB tumors showed a significantly lower risk of radiation necrosis with a Hazard Ratio (HR) of 0.252, *p* < 0.001, independently of the applied radiation dose. Furthermore, higher radiation doses had a significant impact on the occurrence of RN (HR 1.372, *p* = 0.002). Conclusions: The risk for the development of RN for SB tumors appears to be low but should not be underestimated. No difference was found between recurrent tumors and newly diagnosed tumors, which may support the value of radiosurgical treatment for patients with recurrent SB tumors.

## 1. Introduction

Stereotactic Radiosurgery (SRS) is widely used in neuro-oncological patients. The long-term outcomes and local tumor control are comparable to surgical resection [1,2,3,4,5]. Skull base tumors, due to their proximity to crucial anatomical structures, often require a combined management by surgical resection and SRS [6].

Radiation necrosis (RN) is a potential late complication of stereotactic radiosurgery [7,8,9]. The incidence of RN differs between 3.29% and 26.8% for meningiomas and between 0.5% and 12% for vestibular schwannomas [10,11,12,13,14]. Precise prediction of its development is difficult, since only a few risk factors have been demonstrated. Among these are not only larger tumor volumes and high radiation doses [15,16,17] but also specific intracranial locations. In particular, the skull base (SB) has been associated with a reduced risk of post-radiosurgical symptoms in meningiomas [16]. Several options are available for the treatment of RN with bevacizumab shown to be effective in randomized, double-blind trials [18]. In selected cases, surgical treatment may be necessary [19,20].

Particularly in cases of a challenging surgical approach like for SB location or for recurrent tumors, SRS has gained importance. Therefore, it is important to achieve a better understanding of the impact of distinct tumor locations on the occurrence of RN after SRS.

The aim of this study was thus to determine whether tumor location at the SB, regardless of low-malignant tumor type, confers an advantage in postoperative outcomes and a decreased risk of developing RN, as well as to compare it with other known risk factors such as tumor size or radiation dose.

## 2. Materials and Methods

All patients with benign intracranial neoplasms who underwent stereotactic radiosurgery between January 2004 and November 2020 were retrospectively evaluated. Clinical, epidemiological and medication data were gathered from the patients’ electronic medical records. Tumor entity was determined based on a radiological diagnosis (including appearance on MRI or known history of malignancy). Tumor size was defined as largest diameter on axial T1-weighted sequences with gadolinium on pre-interventional MRI and was assessed with an accuracy of 0.1 mm. Radiosurgery was performed with Elekta Synergy and Elekta Precise linear accelerators (Elekta, Stockholm, Sweden), adapted for stereotactic radiosurgery and equipped with interchangeable cone collimators (3–30 mm) using 6 MeV photon beams.

Gadolinium-enhanced T1-weighted volume MR scans were obtained in 3 dimensions less than 72 h prior to treatment in all patients as standard of care for patients undergoing radiosurgical procedures. For the radiosurgery treatment, planning IPlan Software 4.5.8 (BrainLAB AG, Munich, Germany) was used.

Stereotactic radiosurgery was performed with an invasive stereotactic head ring. After placement of the ring and stereotactic localizers, a contrast-enhanced computed tomography (CT) scan was performed. On these CT scans and fused previously obtained MR images, the planning target volume of the tumor and the organs of risk were outlined.

The applied dose was at the discretion of the treating physician and ranged from 12 to 18 Gy to the prescription isodose line (80%). By different combinations of number, span and weight of noncoplanar arcs, high conformality of the treatment dose to the borders of the planning target tumor volume, as well as a steep dose gradient, was established.

As standard of care at the authors’ institution for prevention of RN, patients were routinely administered oral corticosteroids after SRS. Typically, patients were administered 3 × 4 mg dexamethasone for 5 days followed by decreasing dosage of 2 mg every 5 days.

Follow-ups included contrast-enhanced MRI imaging and were performed during the post-treatment course every 3, 6 and 12 months and every 6 to 12 months thereafter until five years after the intervention, with expansion of the intervals after five years. Any changes, especially enlargement, of the pre-treatment tumor volume and surrounding tissue were screened for the occurrence of RN. The distinction to an oncological progression was made according to the clinical and radiological course.

Data on RT-induced complications were also retrospectively analyzed with a special focus on RN defining T1CE + T2 changes suggestive of RN on post-treatment magnetic resonance imaging (MRI). T1CE + T2 changes were defined as any parts of the brain and tumor surrounding tissue with contrast-enhancement and distinct margins best seen on T1 post-contrast sequence, which were often associated with surrounding edema (abnormal T2 hyperintensity). No pathological confirmation for diagnosis of radiation necrosis was necessary due to the mostly radiological diagnosis.

Treatment of RN was based on individual decisions and was mainly conservative, including dexamethasone and—if necessary—bevacizumab, with surgical debulking indicated only in very massive and space-occupying lesions.

The study was conducted following the guidelines of the Declaration of Helsinki and was approved by the Ethics Committee of the Medical University of Innsbruck (1333/2021).

Statistical analysis was processed using IBM SPSS Statistics (IBM SPSS Statistics for Mac OS, Version 26.0. Armonk, NY, USA: IBM Corp.). For analysis, patients were divided into two groups according to the location: SB tumors and non-SB tumors. Normal distribution of scale parameters was determined by the Kolmogorov–Smirnov test. Correlations for non-parametric data were assessed using Spearman’s method. *t*-Tests for normal distributed scale parameters, Mann–Whitney U-test for rank and scale parameters lacking normal distribution and Chi2-test comparing two binominal parameters were assessed according to general terms. The Cox regressions were used to reveal the Hazard ratio for radiation necrosis. The confidence interval (CI) was defined at 95%.

## 3. Results

Overall, 205 patients (120 female, 85 male) could be retrieved from our database and were included in this study. The median age was 58 years with a range from 17 to 88 years (Interquartile Range (IqR) 49–66). The most common tumor types are shown in Table 1. A total of 157 tumors (76.6%) were located at the SB. The neoplasms listed as “other tumors” in Table 1 were predominantly rare intracranial tumor entities, such as choroid plexus papilloma.

The most frequent location for meningiomas was parasagittal, with 24 cases (46%), while 21 (40%) meningiomas were located at the SB.

In this series, the mean diameter on axial imaging was 16.7 ± 5.2 mm (range 5.5–34.8). The median applied radiation dose was 13 Gy (IqR 12–14, range 12–18).

The median cumulative dose of dexamethasone administered following the previously described scheme was 120 mg (IqR 72–120).

For the majority of cases, this was the first radiation therapy to the brain (195 cases, 95.1%). Four patients (1.9%) had already undergone one preceding cranial radiotherapy, and six patients (2.9%) had already undergone two previous cerebral radiation therapies of the same or other intracranial lesion. In 39 patients (19.1%), at least one previous surgical intervention was conducted at the time of SRS.

The mean follow-up amounted to 42 months (Standard deviation ±16.3, range 0–192 months).

In total, 32 patients developed RN during this time (15.6%). Half of the patients required treatment due to neurological symptoms or radiological progression of the RN. Conservative treatment (dexamethasone in 12 cases/Bevacizumab in 2 cases) was administered in the majority of the treated cases (n = 14, 87%); two cases required surgical resection due to a massive space-occupying lesion. The median time to RN was 10 months (IqR 5–12, range 3–29 months). Overall, 47% of our patients harboring RN developed a symptomatic RN, and the remaining 53% were limited to radiological findings. The incidence of RN depending on the tumor type is shown in Table 2. 

The baseline characteristics like age (*p* = n.s.), gender (*p* = n.s.) and tumor diameter (*p* = n.s.) showed no significant difference between patients with SB tumors and tumors in other locations.

Cox regression between patients with tumor location at the SB and those with tumors in other locations, in terms of RN development, showed a significant difference in favor of patients with SB tumors. This analysis revealed a hazard ratio (HR) of 0.139 (CI 0.067–0.284) of SB tumors versus non-SB tumors for the occurrence of radiation necrosis, *p* < 0.001, model *p* < 0.001. Various demographic factors like age and gender, as well as previous treatments, showed no significant impact on the development of RN in our Cox regression. Further results obtained from the Cox regression are shown in Table 3.

Multivariate Cox regression for location at the SB and applied radiation dose was performed to be able to find the influence of radiation dose on the occurrence of RN, where, however, location at the SB was again found to be a significant factor (HR 0.252, CI 0.112–0.571, *p* < 0.001, model *p* < 0.001), as well as the applied radiation dose (HR 1.372, CI 1.123–1.676, *p* = 0.002, model *p* < 0.001) as independent risk factors.

We did not find any significant differences regarding the types of tumors in the development of radiation necrosis.

Results of Kaplan–Meier analysis are shown in Figure 1.

## 4. Discussion

This study was the first to demonstrate a significant difference in RN development after SRS between neoplasms at the SB and neoplasms of other locations independently of tumor entity. Therefore, SRS for SB tumors could gain even more importance in the coming years. The risk of RN in SB patients appears to be low but should not be underestimated. With the increasing incidence of intracranial neoplasia [21,22], the importance of stereotactic radiosurgery is expected to increase. Moreover, with increasing median age in developed countries, the number of SRS for SB tumors will increase as mainly older patients opt for SRS as the less invasive alternative to surgical resection in a challenging surgical area. Despite promising long-term outcomes regarding tumor control, up to one in six patients suffers from radiation necrosis with possible worsening in QoL [20,23]. Prediction of the development of RN remains difficult, as only a few risk factors are known. Therefore, this study aimed to find differences between neoplasms localized at the SB and neoplasms of other locations with regard to the risk of developing an RN after SRS [16].

The underlying reason for the reduced risk for RN in SB tumors could not be clarified in this study. One possible theory could be the increased capillary permeability and an altered microenvironment. The increased expression of vascular endothelial growth factor (VEGF), which plays an important role in tissue death and the development of necrosis, is routinely expressed in brain tissue but less in bone tissue [24,25,26]. Considering the less surrounding brain tissue of SB tumors compared to intraparenchymal lesions, this could explain the lower risk for RN. This is consistent with data from intracranial arteriovenous malformation [27,28]. However, further prospective studies will be needed to prove this hypothesis.

Another possible hypothesis might be the high attention paid to crucial structures close to the SB, like the brainstem, during the planning for SRS. The focus on best dose optimization, as well as sparing of the surrounding tissue, have already been described in previous studies as important factors in preventing RN [29,30]. This could play a role in the lower incidence of RN in SB tumors not only in our cohort but also in the previous publications [16,31].

Interestingly, no significant differences were shown between pre-treated and newly diagnosed patients. Prior surgical manipulation and resulting scarring, plus a changed microenvironment, seemed to have no effect on the development of RN. Considering the risk of recurrence at the skull base due to often subtotal resection and the higher risks for intraoperative complications in recurrent skull base tumors, SRS should be strongly considered as an alternative in an interdisciplinary setting in case of tumor recurrence [32,33,34].

In this series, no increased risk for RN was shown in patients with SB tumors with prior radiotherapy. This confirms the low risk of RN in SB tumors in general since previous radiotherapy has been associated with an increased risk for RN in different tumors in case of re-irradiation [16,23,35]. However, due to the low patient number with re-irradiation in this series, future prospective trials will be needed to better address the role of previous irradiation on the occurrence of RN in SB tumors.

In this study, age was not associated with an increased risk of radiation necrosis. This holds future clinical relevance, as the median age of the population in developed countries will increase in the next decades and therefore, older patients with SB tumors will have to be treated more frequently. In addition, older patients with significant co-morbidities tend to opt for SRS as a less invasive method, resulting in an expected increase in SRS in these patients with SB tumors [36,37]. Therefore, the data obtained in this study seem important for future clinical decision-making, as age does not seem to play a role in the development of RN in SB tumors.

In our cohort we found a higher incidence of RN compared to large international trials of different intracranial tumors [38,39,40]. On the one hand, this could be due to the fact that many studies included only symptomatic RN requiring treatment. However, in our study, we also included those limited to imaging-based diagnoses of RN, potentially leading to a higher incidence thereof. On the other hand, due to research in past years [20], we established a MRI protocol including diffusion-weighted sequence (DTI), post-contrast transverse T1-weighted magnetization-prepared rapid gradient-echo sequence (MPRAGE), as well as Perfusion-Weighted Imaging, as recently published by Mangesius et al. [41]. This MRI protocol allowed us to identify RN with higher accuracy and might also lead to a higher incidence of RN. Furthermore, previous studies already showed that using modern imaging techniques, including perfusion weighted imaging and diffusion weighted imaging, radiation necrosis could be distinguished with high accuracy from tumor progression in different intracranial tumor types [42,43,44,45].

This study suffered limitations mainly based on the retrospective and single-center study design. We could not identify an underlying reason for the decreased incidence of RN in skull base tumors. Due to the surgical challenging approach to the SB, the diagnosis of RN was mostly based solely on imaging protocols and criteria. Further prospective studies, including a translational approach to RN, will be needed.

## 5. Conclusions

In this study, we could show a significantly lower risk for the development of radiation necrosis in skull base tumors independent of tumor entity after stereotactic radiosurgery. Our data, therefore, suggest that tumor location does influence the development of radiation necrosis. No increased risk was shown for previously irradiated patients and elderly patients, which has clinical relevance for the future and emphasizes the importance of stereotactic radiosurgery as an interdisciplinary treatment concept in neuro-oncology.

## Figures and Tables

**Figure 1 cancers-15-04760-f001:**
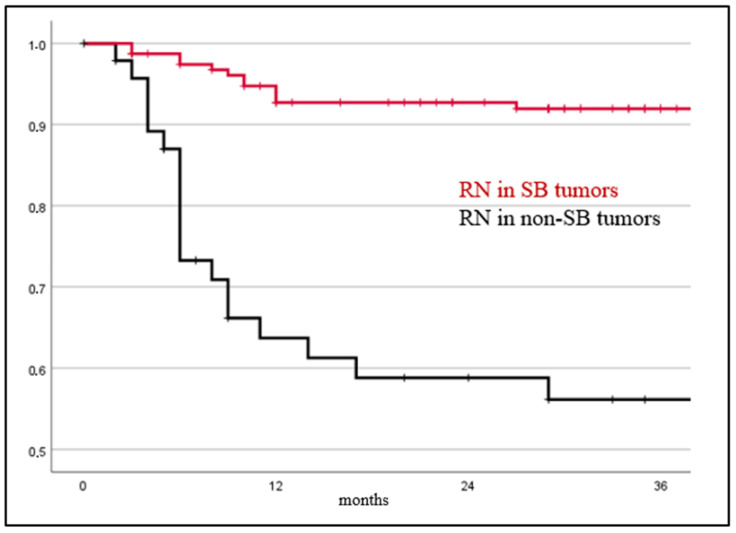
Tumors located at the SB showed a significantly lower risk for developing radiation necrosis in Kaplan–Meier analysis compared to tumors located elsewhere (*p* < 0.001).

**Table 1 cancers-15-04760-t001:** Type of tumors treated with SRS and tumors located at the SB.

	Cases	Percent	SB Location
Vestibular schwannoma	125	60.9	125 (100%)
Meningioma	52	25.4	21 (40.4%)
Glomus jugulare tumors (GJTs)	6	2.9	6 (100%)
Ependymoma	8	3.9	0 (0%)
Schwannoma of other cranial nerves	5	2.4	5 (100%)
Others	9	4.4	0 (0%)

**Table 2 cancers-15-04760-t002:** Incidence of RN in histological tumor types.

	Cases (SB Location)	Radiation Necrosis	No Radiation Necrosis
Vestibular schwannoma	125 (100%)	10 (8%)	115 (92%)
Meningioma	52 (40.4%)	20 (38.5%)	32 (61.5%)
Glomus jugulare tumors (GJTs)	6 (100%)	0 (0%)	6 (100%)
Ependymoma	8 (0%)	1 (12.5%)	7 (87.5%)
Schwannoma of other cranial nerves	5 (100%)	0 (0%)	5 (100%)
Others	9 (0%)	1 (11.1%)	8 (88.9%)

**Table 3 cancers-15-04760-t003:** Location at the SB, but not previous resection or SRS showed a significant impact on the development of RN. Moreover, older patients showed no increased risk for RN.

Risk Factors for RN		HR	Cl	*p*-Value
Location at skull base	yes/no	0.139	0.068–0.284	<0.001
Previous resection	yes/no	1.153	0.950–1.400	0.150
Applied radiation dose	per gray	1.571	1.342–1.840	<0.001
Age	per year	1.009	0.984–1.035	0.479
Previous SRS	yes/no	1.281	0.665–2.467	0.459
Dexamethasone dose	per mg	0.995	0.985–1.004	0.249
Tumor diameter	per mm	1.052	0.990–1.117	0.104
Gender	male/female	1.161	0.567–2.374	0.683

## Data Availability

The datasets generated during and/or analyzed during the current study are available from the corresponding author on reasonable request.

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
