# Peer review of "Tumor Location Impacts the Development of Radiation Necrosis in Benign Intracranial Tumors"

_cancers, 2023, doi:10.3390/cancers15194760_

Round 1
Reviewer 1 Report
To accept this manuscript, authors should describe and discuss the results of radiation necrosis by the type of tumour, because neoplasms of skull base location are different to non-skull base location (for example, the schwannomas).
Authors should show the percentage of radiation necrosis in skull base and non-skull base location and in each histological type of tumour.
In conclusions should respond to the main aim of the study.
Well-written english
Author Response
Dear reviewer,
we are very grateful for your time and consideration reviewing our manuscript.
Without any doubts, your comments and suggestions helped us to enhance the manuscript and clarify results and conclusion.
Please find our responses and comments in this letter and allow us to respond in a point-by-point fashion.
To accept this manuscript, authors should describe and discuss the results of radiation necrosis by the type of tumour, because neoplasms of skull base location are different to non-skull base location (for example, the schwannomas).
Authors should show the percentage of radiation necrosis in skull base and non-skull base location and in each histological type of tumour.
Thank you for your input. We added a table with the percentage of radiation necrosis for each type of tumor to the results.
In conclusions should respond to the main aim of the study.
Thank you for your comment. We extended the conclusion of our manuscript with regard to the main aim of our study.

Reviewer 2 Report
The study investigates the correlation between tumor location and the development of radiation necrosis (RN) following stereotactic radio-surgery (SRS) for benign neoplasms between 2004 and 2020. The primary aim is to differentiate the risk between tumors located at the skull base (SB) and those in non-skull base locations.
Methods:
A retrospective evaluation of 205 patients was conducted.
Clinical, imaging, and medication data were collected.
The largest axial tumor diameter was determined using MRI scans in T1 weighted imaging with gadolinium.
The diagnosis of RN was based on imaging parameters.
Results:
Out of 205 patients, 157 tumors (76.6%) were at the SB, and 48 (23.4%) were non-SB tumors.
Among SB-tumors, vestibular schwannomas and meningiomas were the most common (125 and 21 cases, respectively).
32 patients (15.6%) developed RN after a median of 10 months.
62 patients (30.2%) had undergone at least one surgical resection.
In multivariate Cox regression, SB-tumors had a significantly lower risk of RN (Hazard Ratio of 0.252, p<0.001), independent of the applied radiation dose.
Higher radiation doses significantly impacted the occurrence of RN (HR 1.372, p=0.002).
The study concluded that the risk for the development of RN in SB tumors is low but not negligible. No significant difference was found between recurrent tumors and newly diagnosed tumors, supporting the value of radiosurgical treatment for patients with recurrent SB tumors.
The manuscript has several strengths, including the approach includes extensive clinical, epidemiological, and medication data, as well as MRI, CT scans, and treatment details, allowing for a multifaceted analysis.The use of standardized methods, equipment, and guidelines ensures consistency in treatment and evaluation. Tumor size is assessed with accuracy, and the applied dose is specifically delineated. This adds precision to the evaluation of treatment outcomes. Including the assessment of RT-induced complications like radiation necrosis (RN), adds depth to the analysis of treatment outcomes.
However, the manuscript has several weaknesses. The absence of pathological confirmation for the diagnosis of radiation necrosis may question the accuracy of the diagnosis based solely on radiological findings. Histological diagnosis is considered the gold standard for diagnosing RN, and its absence might raise questions about diagnostic accuracy. Treatment of RN is based on individual decisions may introduce variability in treatment response and make it challenging to draw firm conclusions regarding the best approach for managing RN. The method does not employ a specific rating scale to assess the likelihood of RN, possibly leading to a lack of consistency or standardization in identifying and grading RN. There is no mention of stratifying or adjusting for potential confounding factors like underlying health conditions, previous treatments, or demographic differences, which may influence outcomes. Moreover, the imaging Protocol method relies on MRI with gadolinium enhancement, it does not mention the use of various weighted imaging or dynamic contrast studies, potentially limiting the detail and insights that could be derived from the imaging. A table containing detailed characteristics of treated lesions could benefit the readers.
Author Response
Dear reviewer,
we are very grateful for your time and consideration reviewing our manuscript.
Without any doubts, your comments and suggestions helped us to enhance the manuscript and clarify results and conclusion.
Please find our responses and comments in this letter and allow us to respond in a point-by-point fashion.
The study investigates the correlation between tumor location and the development of radiation necrosis (RN) following stereotactic radio-surgery (SRS) for benign neoplasms between 2004 and 2020. The primary aim is to differentiate the risk between tumors located at the skull base (SB) and those in non-skull base locations.
Methods:
A retrospective evaluation of 205 patients was conducted.
Clinical, imaging, and medication data were collected.
The largest axial tumor diameter was determined using MRI scans in T1 weighted imaging with gadolinium.
The diagnosis of RN was based on imaging parameters.
Results:
Out of 205 patients, 157 tumors (76.6%) were at the SB, and 48 (23.4%) were non-SB tumors.
Among SB-tumors, vestibular schwannomas and meningiomas were the most common (125 and 21 cases, respectively).
32 patients (15.6%) developed RN after a median of 10 months.
62 patients (30.2%) had undergone at least one surgical resection.
In multivariate Cox regression, SB-tumors had a significantly lower risk of RN (Hazard Ratio of 0.252, p<0.001), independent of the applied radiation dose.
Higher radiation doses significantly impacted the occurrence of RN (HR 1.372, p=0.002).
The study concluded that the risk for the development of RN in SB tumors is low but not negligible. No significant difference was found between recurrent tumors and newly diagnosed tumors, supporting the value of radiosurgical treatment for patients with recurrent SB tumors.
The manuscript has several strengths, including the approach includes extensive clinical, epidemiological, and medication data, as well as MRI, CT scans, and treatment details, allowing for a multifaceted analysis. The use of standardized methods, equipment, and guidelines ensures consistency in treatment and evaluation. Tumor size is assessed with accuracy, and the applied dose is specifically delineated. This adds precision to the evaluation of treatment outcomes. Including the assessment of RT-induced complications like radiation necrosis (RN), adds depth to the analysis of treatment outcomes.
Thank you for your comment.
However, the manuscript has several weaknesses. The absence of pathological confirmation for the diagnosis of radiation necrosis may question the accuracy of the diagnosis based solely on radiological findings. Histological diagnosis is considered the gold standard for diagnosing RN, and its absence might raise questions about diagnostic accuracy. Treatment of RN is based on individual decisions may introduce variability in treatment response and make it challenging to draw firm conclusions regarding the best approach for managing RN. The method does not employ a specific rating scale to assess the likelihood of RN, possibly leading to a lack of consistency or standardization in identifying and grading RN. There is no mention of stratifying or adjusting for potential confounding factors like underlying health conditions, previous treatments, or demographic differences, which may influence outcomes. Moreover, the imaging Protocol method relies on MRI with gadolinium enhancement, it does not mention the use of various weighted imaging or dynamic contrast studies, potentially limiting the detail and insights that could be derived from the imaging. A table containing detailed characteristics of treated lesions could benefit the readers.
Thank you for your input. Due to the challenging surgical approach to the skull base, histological verification of suspected radiation necrosis was not possible for all of our patients. We have acknowledged this limitation in our study. However, despite presenting a challenge to the treatment team, several studies by Carr et al, Vellayappan et al, Piper et al, and Furuse et al demonstrated that modern imaging techniques, including perfusion-weighted imaging and diffusion-weighted imaging, can accurately distinguish radiation necrosis from tumor progression in various types of intracranial tumors. We included these publications in our bibliography. As a result, we were able to establish a diagnostic and therapeutic algorithm utilizing advanced imaging at our institution, as previously published in this journal by Mangesius et al. This contribution has been incorporated into our discussion section.
In our Cox regression analysis, we did not observe a significant impact of various demographic factors such as age, gender, and previous treatments on the development of radiation necrosis. The results are presented in table 3. We included this information to provide a clearer explanation of our findings.

Round 2
Reviewer 1 Report
I consider that authors should be show if the percentages of radiation necrosis between the types of tumors are statistically different.
None
